# Clinical characteristics and factors associated with COVID-19-related mortality and hospital admission during the first two epidemic waves in 5 rural provinces in Indonesia: A retrospective cohort study

Henry Surendra[1,2]*, C. Yekti Praptiningsih[3,4], Arina M. Ersanti[3], Mariati Rahmat[3,5], Widia Noviyanti[3], Joshua A. D. Harmani[3], Erni N. A. Mansur[6], Yana Y. Suleman[6], Sitti Sudrani[7], Rosalina Rosalina[8], Ismen Mukhtar[9], Dian Rosadi[10], Lukman Fauzi[11], Iqbal R. F. Elyazar[2], William A. Hawley[4], Hariadi Wibisono[3]

1 Monash University Indonesia, Tangerang Selatan, Indonesia, 2 Oxford University Clinical Research Unit Indonesia, Faculty of Medicine Universitas Indonesia, Jakarta, Indonesia, 3 Perhimpunan Ahli Epidemiologi Indonesia, Jakarta, Indonesia, 4 US Centers for Disease Control and Prevention, Jakarta, Indonesia, 5 Dinas Kesehatan Kabupaten Sinjai, Sinjai, Indonesia, 6 Dinas Kesehatan Provinsi Gorontalo, Gorontalo, Indonesia, 7 Dinas Kesehatan Provinsi Sulawesi Tengah, Manado, Indonesia, 8 Dinas Kesehatan Provinsi Sulawesi Tenggara, Kendari, Indonesia, 9 Dinas Kesehatan Provinsi Lampung, Bandar Lampung, Indonesia, 10 Faculty of Medicine, Universitas Lambung Mangkurat, Banjarmasin, Indonesia, 11 Public Health Science Department, Universitas Negeri Semarang, Semarang, Indonesia

* henrysurendra.15@gmail.com, henry.surendra@monash.edu

## Abstract

### Background

Data on coronavirus disease 2019 (COVID-19) clinical characteristics and severity from resource-limited settings are limited. This study examined clinical characteristics and factors associated with COVID-19 mortality and hospitalisation in rural settings of Indonesia, from 1 January to 31 July, 2021.

### Methods

This retrospective cohort included individuals diagnosed with COVID-19 based on polymerase chain reaction or rapid antigen diagnostic test, from five rural provinces in Indonesia. We extracted demographic and clinical data, including hospitalisation and mortality from a new piloted COVID-19 information system named Sistem Informasi Surveilans Epidemiologi (SISUGI). We used mixed-effect logistic regression to examine factors associated with COVID-19-related mortality and hospitalisation.

### Results

Of 6,583 confirmed cases, 205 (3.1%) died and 1,727 (26.2%) were hospitalised. The median age was 37 years (Interquartile range 26–51), with 825 (12.6%) under 20 years, and 3,371 (51.2%) females. Most cases were symptomatic (4,533; 68.9%); 319 (4.9%) had a

**Data Availability Statement:** The data contain de-identified data set own by the Indonesian Epidemiologist Association/Perhimpunan Ahli Epidemiologi Indonesia. After publication, the datasets used for this study will be made available to others on reasonable requests to the corresponding author and the Indonesian Epidemiologist Association (email: info@paei.or.id), including a detailed research proposal, study objectives and statistical analysis plan.

**Funding:** This study received funding from South Asia Field Epidemiology and Technology Network (SAFETYNET), Grant Number: NU2HGH000042, Strengthening Regional Field Epidemiology Training Program (FETP) Networks. The funders had no role in study design, data collection and analysis, decision to publish, or preparation of the manuscript.

**Competing interests:** The authors have declared that no competing interests exist.

clinical diagnosis of pneumonia and 945 (14.3%) presented with at least one pre-existing comorbidity. Age-specific mortality rates were 0.9% (2/215) for 0–4 years; 0% (0/112) for 5–9 years; 0% (1/498) for 10–19 years; 0.8% (11/1,385) for 20–29 years; 0.9% (12/1,382) for 30–39 years; 2.1% (23/1,095) for 40–49 years; 5.4% (57/1,064) for 50–59 years; 10.8% (62/576) for 60–69 years; 15.9% (37/232) for ≥70 years. Older age, pre-existing diabetes, chronic kidney disease, liver diseases, malignancy, and pneumonia were associated with higher risk of mortality and hospitalisation. Pre-existing hypertension, cardiac diseases, COPD, and immunocompromised condition were associated with risk of hospitalisation but not with mortality. There was no association between province-level density of healthcare workers with mortality and hospitalisation.

## Conclusion

The risk of COVID-19-related mortality and hospitalisation was associated with higher age, pre-existing chronic comorbidities, and clinical pneumonia. The findings highlight the need for prioritising enhanced context-specific public health action to reduce mortality and hospitalisation risk among older and comorbid rural populations.

## Background

The coronavirus disease 2019 (COVID-19) pandemic continues to affect the world population. Globally, there have been more than 760 million total confirmed cases, and 6.8 million deaths reported to the World Health Organization (WHO) as of 15 March 2023 [1]. As per 11 March 2023, over 13 billion COVID-19 vaccine doses have been administered worldwide [1], yet 28,018 new COVID-19-related deaths were still being reported in the last 28 days [2], with remaining challenges in making effective vaccines and anti-viruses against the new strains of SARS-CoV-2 [3].

Current evidence suggests that severe outcomes of COVID-19 have been consistently associated with older age and pre-existing chronic conditions, such as hypertension, diabetes, obesity, cardiac disease, chronic kidney disease, and liver disease [4–11]. Moreover, recent findings from US [12, 13], Chile [14], Brazil [15], and including from Indonesia [16] have suggested that poor outcomes associated with COVID-19 are concentrated in groups with higher socio-demographic and health care vulnerability. However, the understanding of COVID-19 severity mostly comes from studies conducted in urban settings. Limited data exist from rural settings on COVID-19 clinical characteristics and factors associated with severity where differences in age distribution, comorbidities, access to quality health services, and other factors may influence trends regarding severe outcomes [9,16–19].

Indonesia is a low-middle income country (LMIC) that has suffered the highest number of COVID-19 confirmed cases and deaths in Southeast Asia, second only to India in Asia [2]. Since the first reported cases on 2 March 2020, Indonesia has reported a total of more than 6.7 million cases and 160,956 deaths (2.4% confirmed case fatality rate) up to 15 March 2023 [20]. The majority of cases and deaths in Indonesia were reported on Java Island, a more developed setting populated by 152 million individuals (56% of Indonesia's total population). The first epidemic wave occurred from 2 March 2020 to 30 April 2021, and the more intense second wave dominated by Delta variant peaked in July 2021 [20]. Recent studies from Indonesia's capital city of Jakarta suggested that COVID-19 disproportionately affected individuals with

older age and pre-existing chronic comorbidities, as well as those areas with lower vaccine coverage and higher poverty and population density [9, 16]. However, data on the impact of COVID-19 in less developed areas of Indonesia with greater geographic, cultural, and socio-economic diversity, outside of the island of Java, are scarce.

The limited evidence regarding COVID-19 epidemiology and severity from low resource settings in Indonesia–especially in areas outside of Java–has been particularly associated with the lack of epidemiological data recorded by the surveillance systems in these areas. The Indonesian government has been implementing a national COVID-19 data collection system (New All Record) that collect limited demographic data (sex, age, and residential) of individuals tested positive by either polymerase chain reaction (PCR) or rapid antigen diagnostic test (Ag RDT) [21, 22], followed by the Peduli Lindungi [23], a system that records data on vaccination status, latest PCR/Ag RDT results, and travel history. To improve quality of COVID-19 surveillance and response in Indonesia, the Indonesian Epidemiologist Association (Perhimpunan Ahli Epidemiologi Indonesia, PAEI) in collaboration with the Ministry of Health of Indonesia implemented a new COVID-19 information system named Sistem Informasi Surveilans Epidemiologi (SISUGI) [24] to collect more detailed clinical and epidemiological data in five rural provinces–East Nusa Tenggara in eastern Indonesia, Gorontalo, Central Sulawesi, and Southeast Sulawesi on the island of Sulawesi, and Lampung on the island of Sumatra (Fig 1). This study utilised the epidemiological surveillance data recorded in the SISUGI, to better understand clinical characteristics and factors associated with COVID-19-related mortality and hospitalisation in five rural provinces in Indonesia, from 1 January to 31 July 2021. The

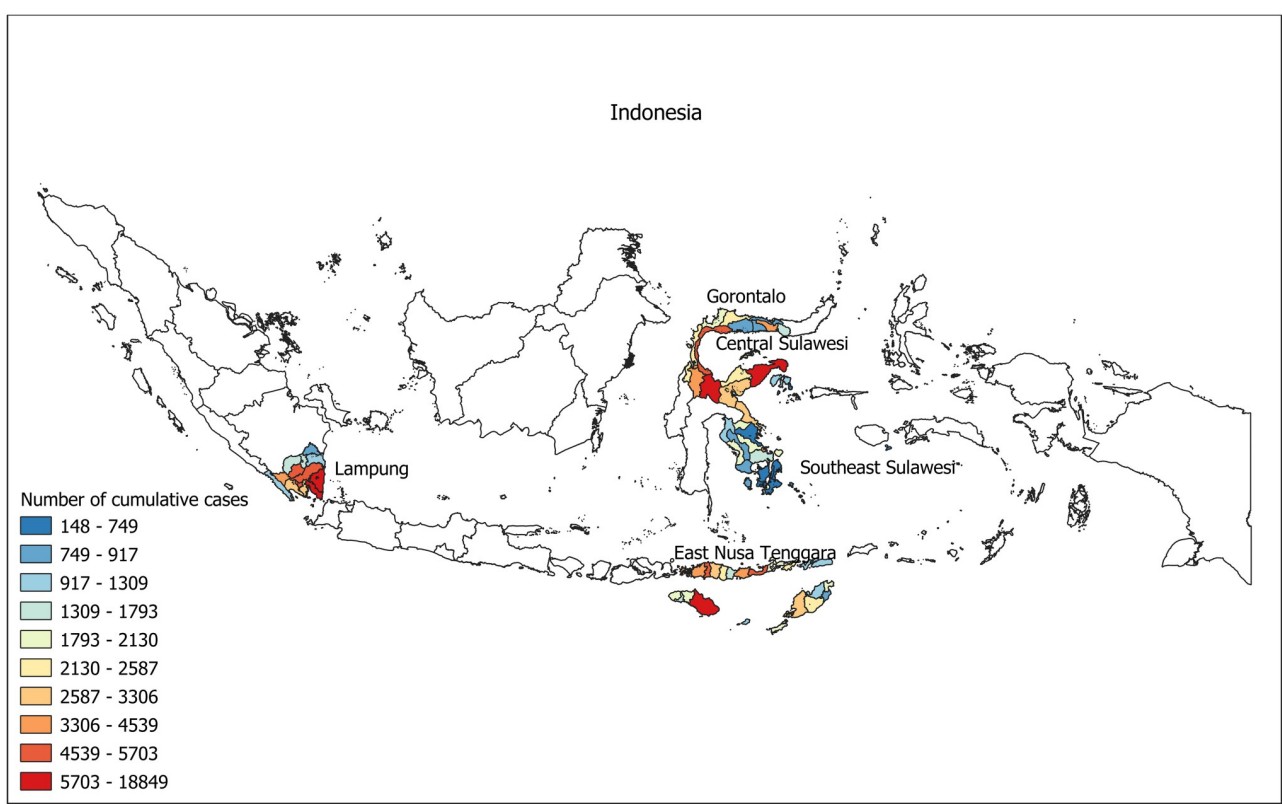

**Fig 1. Map showing five study sites in East Nusa Tenggara, Gorontalo, Central Sulawesi, Southeast Sulawesi, and Lampung province, and spatial distribution of cumulative COVID-19 cases by district.** Map was obtained from the Indonesia Geospatial Agency; https://tanahair.indonesia.go.id/portal-web/. COVID-19 aggregated data were obtained from https://data.covid19.go.id/.

study findings will inform decision on prioritising clinical and public health interventions to reduce risk of hospital admission and mortality associated with COVID-19 in rural settings of Indonesia.

## Materials and methods

### Study design and participants

This was a retrospective cohort analysis to assess demographic and clinical factors associated with mortality and hospitalisation of adults and children diagnosed with COVID-19 in five provinces in Indonesia. The study population included individuals diagnosed with COVID-19 based on either a PCR test or Ag RDT done in either health facilities or laboratory from Lampung, Gorontalo, Central Sulawesi, Southeast Sulawesi, and East Nusa Tenggara provinces between January to July 2021. In accordance with Indonesia's national COVID-19 guidelines, individuals are categorized as COVID-19 confirmed cases if they tested positive by Ag RDT or PCR [25]. Based on data from Indonesia Central Bureau of Statistics, the majority of areas (villages) in the five provinces were categorised as rural areas i.e., Lampung (68%), Gorontalo (66%), Central Sulawesi (55%), Southeast Sulawesi (83%), and East Nusa Tenggara (92%) [26, 27]. Based on the number of doctors, nurses, midwives, and public health officer per 100,000 population (S1 Table), these five provinces were considered as rural settings with limited health care capacity. The five provinces were purposively selected as pilot project sites for the implementation of SISUGI developed by PAEI in collaboration with the US CDC Indonesia Country Office, South Asia Field Epidemiology and Technology Network (SAFETYNET), and Ministry of Health of the Republic of Indonesia. This study was conducted as part of a program evaluation.

### Data collection

Each health facility had a surveillance officer responsible for conducting an epidemiological investigation of each confirmed case via interview using a standard form. Completed forms were submitted to the SISUGI database. Data regarding dates of test, onset of illness, SARS-CoV-2 Ag RDT and PCR testing, and outcomes (mortality and hospital admission) were recorded, along with age, gender, symptoms, and comorbidities. Comorbidities were recorded based on clinical assessments at admission or patient reporting. Fever was defined as axillary temperature of $\geq 38\,^{\circ}$C. Clinical diagnosis of pneumonia (indicative of moderate or severe disease [28]) was determined by the treating physician, based on clinical and radiological evaluation [29]. Individual-level epidemiological data of all COVID-19 confirmed cases between January to July 2021 were then extracted from the SISUGI database. Province-level data were collected from various government sources. Data on population numbers by province in 2020 were collected from the Statistics Bureau Database [30]. Data on the number of doctors, nurses, midwives per province, as per October 2021 were collected from the Indonesia Ministry of Health records [31].

### Statistical analysis

Descriptive statistics included proportions for categorical variables and medians and interquartile ranges (IQRs) for continuous variables. We used the Mann-Whitney U test to compare median different of age between deceased and recovered cases, and between hospitalised and non-hospitalised cases. For categorical variables, we used Chi-squared test, or Fisher's exact test to compare other characteristics between deceased and recovered cases, and between

hospitalised and non-hospitalised cases. Due to the high proportion of missing data needed to define the time from onset to admission, this variable was not analysed.

We used bivariable and multivariable mixed effects logistic regression models to determine the risk of death and hospitalisation, expressed as odds ratios with 95% confidence intervals. Province was treated as the random effect variable to adjust for clustering of observations within provinces. We did null model analysis (no predictor was added) and the result justified the use of the mixed effects models. All independent variables with p-value <0.10 in bivariable analysis were included in the multivariable models. Sex was treated as a "forced variable" (a strong potential confounder to be included in multivariable models, regardless of its p-value in bivariable analysis). Final model selection was informed by likelihood ratio tests. We reported two final multivariable logistic regression models. In the first model, we assessed the effect of each type of comorbidity (hypertension, diabetes, cardiac disease, chronic obstructive pulmonary disease (COPD), chronic kidney disease, immunocompromised status, and liver disease), along with other demographic and clinical factors. In the second model, we assessed the effect of the number of comorbidities (0, 1 or >1), controlling for other demographic and clinical factors. We used interaction terms to examine potential effect modification by age and sex. We set statistical significance at 0.05, and all tests were two sided. All analyses were done in Stata/IC 15.1 (StataCorp, College Station, TX, USA). This study is reported as per Strengthening the Reporting of Observational Studies in Epidemiology (STROBE) guidelines [32].

## Ethics approval

This study was approved by the Health Research Ethics Committee of Universitas Negeri Semarang (300/2/KEPK/EC/2022). The requirement for patient consent was waived as this was a secondary analysis of anonymised routine surveillance data.

## Results

### General characteristics of study participants

Between 1 January and 31 July 2021, a total of 6,583 confirmed COVID-19 cases were recorded on the SISUGI. The completeness of data was high, with less than 1% missing data across the variables analysed in the study. The study flow chart and completeness of data are presented in Fig 2.

Table 1 presents the demographic and clinical characteristics of the 6,583 cases included in the analysis. The highest proportion of cases were reported in Lampung (32.6%, 2,146/6,583), and the lowest was in East Nusa Tenggara (10.0%, 658/6,583). The median age was 37 years (IQR 26–51, range 0.1–96), with 825 (12.6%) under 20 years, and 3,371 (51.2%) were females. The incidence rate ranged from 12 per 100,000 populations in Lampung to 116 per 100,000 populations in Gorontalo (S2 Table).

### Clinical characteristics and COVID-19-related mortality and hospitalisation

Of 6,583 cases included in the analysis, 205 (3.1%) were deceased, and 1,727 (26.2%) were hospitalised. The number of COVID-19-related deaths and hospital admissions varied over time and by province (S1 Fig). There were 4,533 (68.9%) symptomatic cases, with the most common presenting symptom being cough (46.7%, 3,072), followed by fever (44.0%, 2,894), runny nose (33.0%, 2,174), malaise (25.4%, 1,671), and others (Table 1 and Fig 3A). The median number of symptoms was 2 (IQR 0–4, range 1–13). There were 319 (4.9%) patients with a clinical diagnosis of pneumonia. Overall, 14.3% (945) of cases had a record of one or more pre-existing

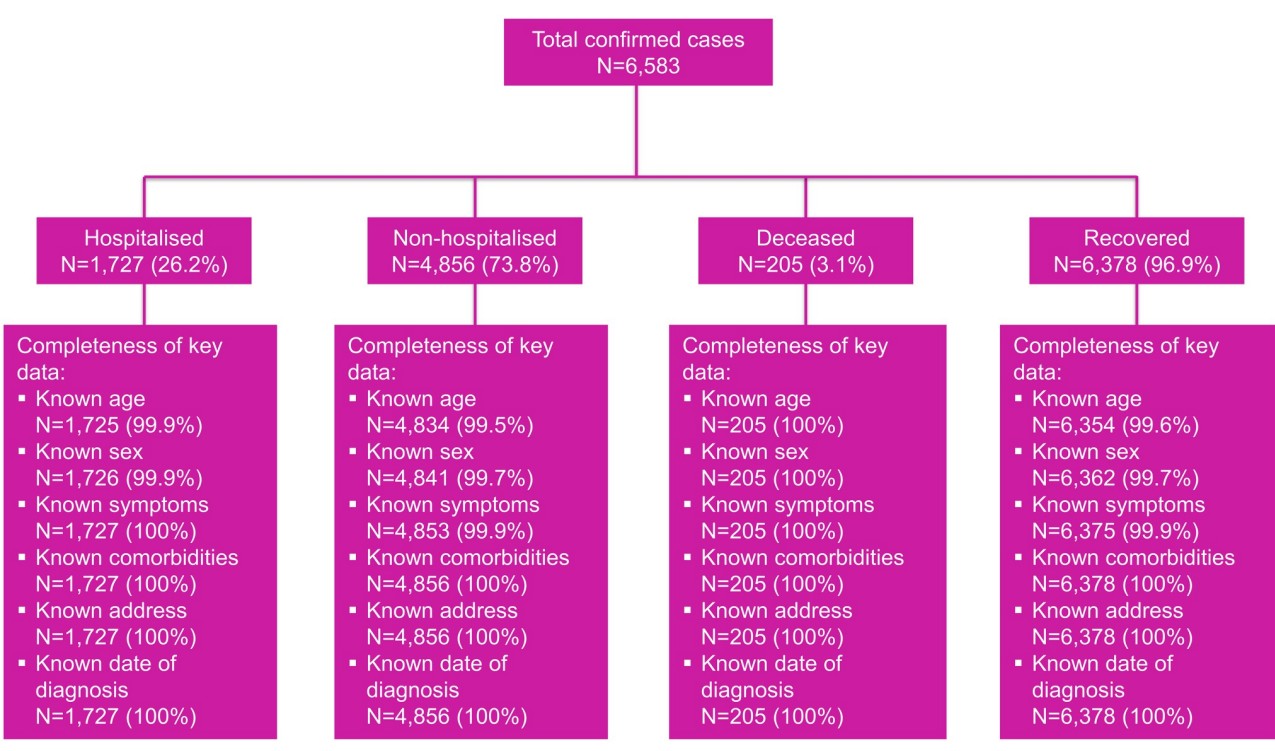

**Fig 2. Study flow chart and completeness of key variables.**

comorbidities, including hypertension (8.2%, 539), diabetes mellitus (5.5%, 365), cardiac disease (2.8%, 187), and others (Table 1 and Fig 3B).

Compared to recovered cases, deceased cases were older (median 58 vs 36 years); more likely to have more presenting symptoms (median 3 vs 2); to have pneumonia (33.2% vs 3.9%); and to have a history of any comorbidity (60.5% vs 12.9%), 1 comorbidity (34.2% vs 10.0%) or >1 comorbidities (26.3% vs 2.9%), specifically hypertension (33.2% vs 7.4%), diabetes (28.8% vs 4.8%), and cardiac disease (15.1% vs 2.5%) (Table 1).

Compared to non-hospitalised cases, hospitalised cases were older (median 50 vs 34 years); more likely to have more presenting symptoms (median 3 vs 1); to have pneumonia (16.2% vs 0.8%); and to have a history of any comorbidity (34.9% vs 7.1%), 1 comorbidity (25.0% vs 5.7%) or >1 comorbidities (9.9% vs 1.4%), specifically hypertension (19.2% vs 4.3%), diabetes (14.3% vs 2.4%), and cardiac disease (7.1% vs 1,3%) (Table 1).

The mortality and hospitalisation rate increased with age (Table 2). Whilst age-specific mortality rate ranged from 0.9% (2/215) for 0–4 years to 15.9% (37/232) for ≥70 years, the age-specific hospitalisation rate ranged from 17.2% (37/215) for 0–4 years to 56.5% (131/232) for ≥70 years. Based on province, the mortality and hospitalisation rate ranged from 1.9% (13/658) and 13.4% (88/658) in East Nusa Tenggara to 4.3% (93/2,146) and 36.1% (775/2,146) in Lampung. Details on province and age-specific mortality and hospitalisation rate can be seen in Table 2.

## Factors associated with COVID-19-related mortality and hospitalisation

In bivariable mixed effects logistic regression analysis (S3 Table), the risk of death was associated with older age, male sex, clinical diagnosis with pneumonia, number of pre-existing comorbidities, hypertension, diabetes, cardiac disease, COPD, chronic kidney disease, liver

**Table 1. Demographics, clinical characteristics, health care workers, and COVID-19-related mortality and hospitalisation in 5 rural provinces in Indonesia, 1 January to 31 July 2021.**

| | Total N = 6,583* | Deceased | | | Hospitalised | | |
|---|---|---|---|---|---|---|---|
| | | Yes (N = 205) | No (N = 6,378) | p value | Yes (N = 1,727) | No (N = 4,856) | p value |
| **Demographics** | | | | | | | |
| Province | | | | 0.002 | | | <0.001 |
| Lampung | 2,146 (32.6) | 93 (45.4%) | 2,053 (32.2%) | | 775 (44.9%) | 1,371 (28.2%) | |
| Gorontalo | 1,356 (20.6%) | 34 (16.6%) | 1,322 (20.7%) | | 340 (19.7%) | 1,016 (20.9%) | |
| Central Sulawesi | 987 (15.0%) | 27 (13.2%) | 960 (15.1%) | | 258 (14.9%) | 729 (15.0%) | |
| Southeast Sulawesi | 1,436 (21.8%) | 38 (18.5%) | 1,398 (21.9%) | | 266 (15.4%) | 1,170 (24.1%) | |
| East Nusa Tenggara | 658 (10.0%) | 13 (6.3%) | 645 (10.1%) | | 88 (5.1%) | 570 (11.7%) | |
| Median age (IQR), years | 37 (26–51) | 58 (50–68) | 36 (26–51) | <0.001 | 50 (33–59) | 34 (25–47) | <0.001 |
| Age group, years | | | | <0.001 | | | <0.001 |
| 0–4 | 215 (3.3%) | 2 (1.0%) | 213 (3.3%) | | 37 (2.1%) | 178 (3.7%) | |
| 5–9 | 112 (1.7%) | 0 (0.0%) | 112 (1.8%) | | 14 (0.8%) | 98 (2.0%) | |
| 10–19 | 498 (7.6) | 1 (0.5%) | 497 (7.8%) | | 65 (3.8%) | 433 (8.9%) | |
| 20–29 | 1,385 (21.0%) | 11 (5.4%) | 1,374 (21.5%) | | 213 (12.3%) | 1,172 (24.1%) | |
| 30–39 | 1,382 (21.0) | 12 (5.9%) | 1,370 (21.5%) | | 249 (14.4%) | 1,133 (23.3%) | |
| 40–49 | 1,095 (16.6) | 23 (11.2%) | 1,072 (16.8%) | | 273 (15.8%) | 822 (16.9%) | |
| 50–59 | 1,064 (16.2%) | 57 (27.8%) | 1,007 (15.8%) | | 446 (25.8%) | 618 (12.7%) | |
| 60–69 | 576 (8.8%) | 62 (30.2%) | 514 (8.1%) | | 297 (17.2%) | 279 (5.8%) | |
| ≥70 | 232 (3.5%) | 37 (18.1%) | 195 (3.1%) | | 131 (7.6%) | 101 (2.1%) | |
| Unknown | 24 (0.4%) | 0 (0.0%) | 24 (0.4%) | | 2 (0.1%) | 22 (0.5%) | |
| Gender | | | | 0.074 | | | 0.115 |
| Female | 3,371 (51.2%) | 90 (43.9%) | 3,281 (51.4%) | | 904 (52.4%) | 2,467 (50.8%) | |
| Male | 3,196 (48.6%) | 115 (56.1%) | 3,081 (48.3%) | | 822 (47.6%) | 2,374 (48.9%) | |
| Unknown | 16 (0.2%) | 0 (0.0%) | 16 (0.3%) | | 1 (0.1%) | 15 (0.3%) | |
| **Clinical characteristics** | | | | | | | |
| Present with any symptoms | 4,533 (68.9%) | 178 (86.8%) | 4,355 (68.3%) | <0.001 | 1,484 (85.9%) | 3,049 (62.8%) | <0.001 |
| Presenting symptoms | | | | | | | |
| Cough | 3,072 (46.7%) | 114 (55.6%) | 2,958 (46.4%) | 0.032 | 975 (56.5%) | 2,097 (43.2%) | <0.001 |
| Fever | 2,894 (44.0%) | 102 (49.8%) | 2,792 (43.8%) | 0.227 | 1,017 (58.9%) | 1,877 (38.7%) | <0.001 |
| Runny nose | 2,174 (33.0%) | 42 (20.5%) | 2,132 (33.4%) | 0.001 | 453 (26.2%) | 1,721 (35.4%) | <0.001 |
| Malaise | 1,671 (25.4%) | 114 (55.6%) | 1,557 (24.4%) | 0.000 | 770 (44.6%) | 901 (18.6%) | <0.001 |
| Headache | 1,447 (22.0%) | 52 (25.4%) | 1,395 (21.9%) | 0.472 | 471 (27.3%) | 976 (20.1%) | <0.001 |
| Sore throat | 1,072 (16.3%) | 35 (17.1%) | 170 (82.9%) | 0.909 | 286 (16.6%) | 786 (16.2%) | 0.551 |
| Myalgia | 1,030 (15.7%) | 48 (23.4%) | 982 (15.4%) | 0.008 | 357 (20.7%) | 673 (13.9%) | <0.001 |
| Anosmia | 942 (14.3%) | 12 (5.9%) | 930 (14.6%) | <0.001 | 202 (11.7%) | 740 (15.2%) | <0.001 |
| Shortness of breath | 858 (13.0%) | 121 (59.0%) | 737 (11.7%) | <0.001 | 556 (32.2%) | 302 (6.2%) | <0.001 |
| Nausea/vomiting | 755 (11.5%) | 53 (25.9%) | 702 (11.0%) | <0.001 | 410 (23.7%) | 345 (7.1%) | <0.001 |
| Ageusia | 321 (4.9%) | 10 (4.9%) | 311 (4.9%) | 0.006 | 72 (4.2%) | 249 (5.1%) | <0.001 |
| Abdominal pain | 343 (5.2%) | 29 (14.2%) | 314 (4.9%) | <0.001 | 189 (10.9%) | 154 (3.2%) | <0.001 |
| Diarrhoea | 253 (3.8%) | 11 (5.4%) | 242 (3.8%) | 0.491 | 104 (6.0%) | 149 (3.1%) | <0.001 |
| Number of symptoms, median (IQR) | 2 (0–4) | 3 (1–5) | 2 (0–4) | <0.001 | 3 (1–5) | 1 (0–4) | <0.001 |
| Clinical diagnosis with pneumonia | 319 (4.9%) | 68 (33.2%) | 251 (3.9%) | <0.001 | 279 (16.2%) | 40 (0.8%) | <0.001 |
| Number of comorbidities | | | | <0.001 | | | <0.001 |
| 0 | 5,638 (85.6%) | 81 (39.5%) | 5,557 (87.1%) | | 1,124 (65.1%) | 4,514 (93.9%) | |
| 1 | 707 (10.7%) | 70 (34.2%) | 637 (10.0%) | | 432 (25.0%) | 275 (5.7%) | |
| >1 | 238 (3.6%) | 54 (26.3%) | 184 (2.9%) | | 171 (9.9%) | 67 (1.4%) | |

*(Continued)*

**Table 1.** (Continued)

| | Total N = 6,583* | Deceased | | | Hospitalised | | |
|---|---|---|---|---|---|---|---|
| | | Yes (N = 205) | No (N = 6,378) | p value | Yes (N = 1,727) | No (N = 4,856) | p value |
| Type of comorbidity | | | | | | | |
| Hypertension | 539 (8.2%) | 68 (33.2%) | 471 (7.4%) | <0.001 | 332 (19.2%) | 207 (4.3%) | <0.001 |
| Diabetes | 365 (5.5%) | 59 (28.8%) | 306 (4.8%) | <0.001 | 247 (14.3%) | 118 (2.4%) | <0.001 |
| Cardiac diseases | 187 (2.8%) | 31 (15.1%) | 156 (2.5%) | <0.001 | 122 (7.1%) | 65 (1.3%) | <0.001 |
| COPD | 48 (0.7%) | 11 (5.4%) | 37 (0.6%) | <0.001 | 41 (2.4%) | 7 (0.1%) | <0.001 |
| Chronic kidney diseases | 46 (0.7%) | 16 (7.8%) | 30 (0.5%) | <0.001 | 36 (2.1%) | 10 (0.2%) | <0.001 |
| Malignancy | 20 (0.3%) | 4 (2.0%) | 16 (0.3%) | <0.001 | 11 (0.6%) | 9 (0.2%) | 0.003 |
| Immunocompromised | 17 (0.3%) | 0 (0.0%) | 17 (0.3%) | 0.459 | 11 (0.6%) | 6 (0.1%) | <0.001 |
| Liver diseases | 14 (0.2%) | 4 (2.0%) | 10 (0.2%) | <0.001 | 11 (0.6%) | 3 (0.1%) | <0.001 |

*5,104 (78%) were PCR-confirmed, and 1,479 (22%) were Ag RDT-confirmed.

IQR: inter quartile range.

diseases, malignancy, lower number of nurses per 100,000 population, and higher number of midwives per 100,000 population. The risk of being hospitalised was associated with older age, clinical diagnosis with pneumonia, number of pre-existing comorbidities, hypertension, diabetes, cardiac disease, COPD, chronic kidney disease, liver diseases, malignancy, immunocompromised, and lower number of nurses per 100,000 populations.

Based on the first multivariable mixed effects logistic regression model (Table 3), the risk of death increased for age groups 50–59 years (aOR 6.5, 95%CI 2.0–21.3), 60–69 years (aOR 10.8, 95%CI 3.2–35.8), and ≥70 years (aOR 18.3, 95%CI 5.3–63.2), compared to <20 years; for cases with pre-existing diabetes (aOR 2.0, 95%CI 1.3–3.2), chronic kidney disease (aOR 6.1, 95%CI 2.5–14.6), liver diseases (aOR 10.3, 95% CI 1.8–58.7), or malignancy (aOR 7.4, 95% CI 1.5–36.9); for patients who had pneumonia (aOR 8.0, 95%CI 5.3–12.0), compared to those without pre-existing comorbidities.

The risk of being hospitalised increased for age groups 40–49 years (aOR 1.7, 95%CI 1.3–2.2), 50–59 years (aOR 2.9, 95%CI 2.2–3.8), 60–69 years (aOR 3.3, 95%CI 2.4–4.4), and ≥70 years (aOR 3.7, 95%CI 2.5–5.5), compared to <20 years; for cases with pre-existing hypertension (aOR 2.4, 95%CI 1.9–3.0), diabetes (aOR 2.2, 95%CI 1.7–2.9), cardiac disease (aOR 2.1, 95% CI 1.4–3.1), COPD (aOR 7.5, 95% CI 2.3–24.1), chronic kidney disease (aOR 3.6, 95%CI 1.6–8.2), liver diseases (aOR 7.5, 95% CI 1.8–31.0), malignancy (aOR 3.4, 95% CI 1.1–10.4), or immunocompromised (aOR 6.9, 95% CI 2.4–20.2); for patients who had pneumonia (aOR 15.3, 95%CI 10.7–21.8).

In the second multivariable model (S4 Table), we found that both the risk of death and hospitalisation increased with the number of pre-existing comorbidities, after controlling for age, sex, pneumonia, and number of health care workers. Sex and number of health care workers at province-level were not associated with COVID-19-related hospitalisation and mortality. Sex and age were not found to be an effect modifier in either model.

## Discussion

This retrospective cohort study described the clinical characteristics and factors associated with risk of COVID-19-related mortality and hospitalisation based on epidemiological surveillance data of confirmed COVID-19 cases reported from five rural provinces of Indonesia, during January to July 2021. This analysis was one of the first COVID-19 epidemiological studies

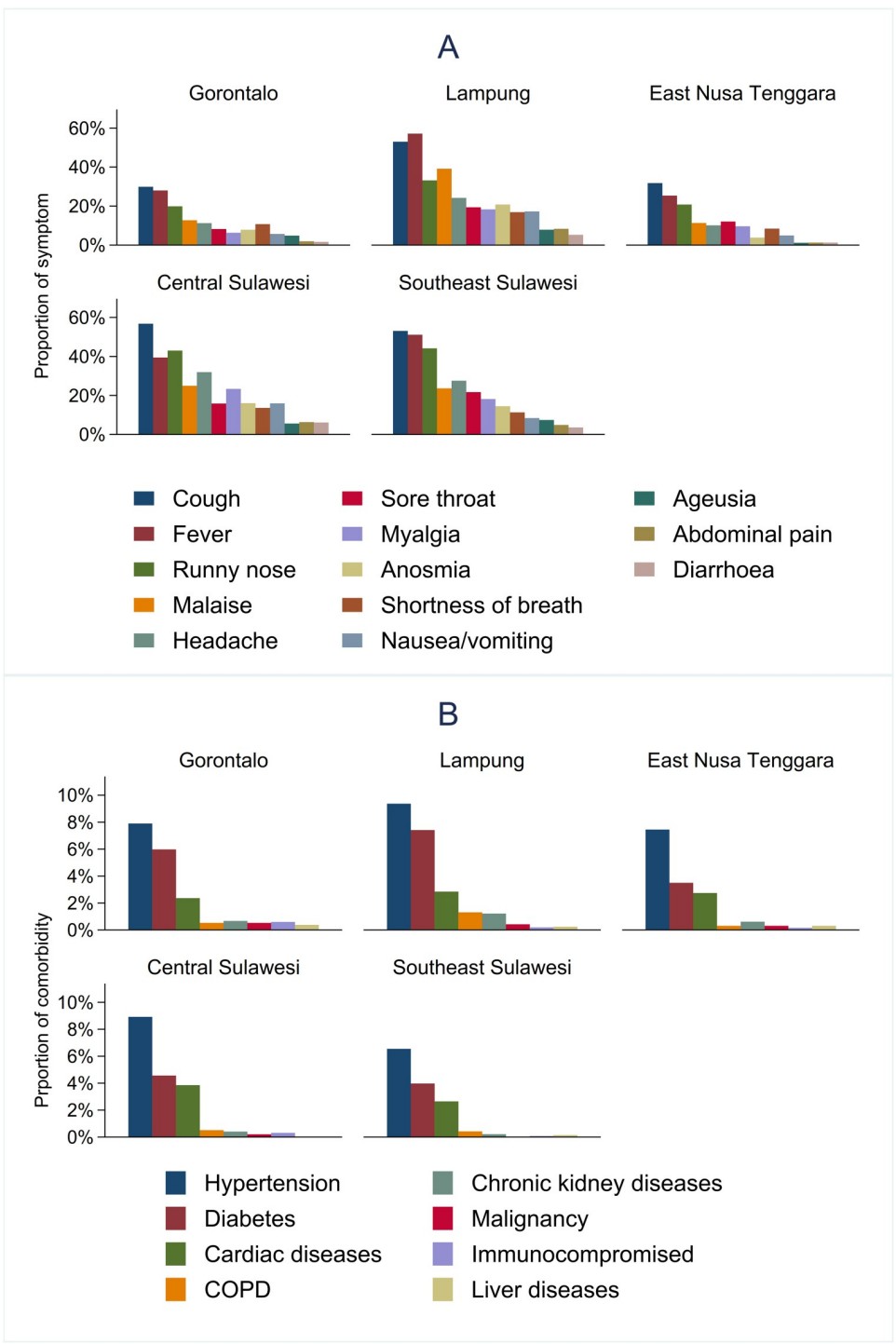

**Fig 3. Presenting symptoms (A) and comorbidity (B) among symptomatic cases in 5 provinces in Indonesia.**

reporting data from rural settings of Indonesia. The observed disease pattern broadly reflects that reported globally, with patients usually presenting multiple symptoms of fever, cough, malaise and/or shortness of breath. The overall mortality and hospitalisation rate was 3.1% (205/6,583) and 26.2% (1,727/6,583), respectively. The age-specific mortality and

**Table 2. Province and age-specific mortality and hospitalisation rate with confidence limits among COVID-19 confirmed cases in five rural provinces in Indonesia, 1 January to 31 July 2021.**

| | N total | Mortality | | Hospitalisation | |
|---|---|---|---|---|---|
| | | N | % (95% CI) | N | % (95% CI) |
| Province | | | | | |
| Lampung | 2,146 | 93 | 4.33 (3.55–5.28) | 775 | 36.1 (34.1–38.2) |
| Gorontalo | 1,356 | 34 | 2.51 (1.80–3.49) | 340 | 25.1 (22.8–27.5) |
| Central Sulawesi | 987 | 27 | 2.74 (1.88–3.96) | 258 | 26.1 (23.5–29.0) |
| Southeast Sulawesi | 1,436 | 38 | 2.65 (1.93–3.62) | 266 | 18.5 (16.6–20.6) |
| East Nusa Tenggara | 658 | 13 | 1.98 (1.15–3.38) | 88 | 13.4 (11.0–16.2) |
| Age group (years) | | | | | |
| 0–4 | 215 | 2 | 0.93 (0.23–3.64) | 37 | 17.2 (12.7–22.9) |
| 5–9 | 112 | 0 | NA | 14 | 12.5 (7.5–20.0) |
| 10–19 | 498 | 1 | 0.20 (0.03–1.41) | 65 | 13.1 (10.4–16.3) |
| 20–29 | 1,385 | 11 | 0.79 (0.44–1.43) | 213 | 15.4 (13.6–17.4) |
| 30–39 | 1,382 | 12 | 0.87 (0.49–1.52) | 249 | 18.0 (16.1–20.1) |
| 40–49 | 1,095 | 23 | 2.10 (1.40–3.14) | 273 | 24.9 (22.5–27.6) |
| 50–59 | 1,064 | 57 | 5.36 (4.15–6.88) | 446 | 41.9 (39.0–44.9) |
| 60–69 | 576 | 62 | 10.76 (8.48–13.57) | 297 | 51.6 (47.5–55.6) |
| ≥70 | 232 | 37 | 15.95 (11.78–21.24) | 131 | 56.5 (50.0–62.7) |

NA: Not Applicable due to 0 value of observation among deceased cases aged 5–9 years old.

hospitalisation rate for children under five years old was 0.9% (2/215) and 17.2% (37/215), respectively. The mortality and hospitalisation rate were highly heterogeneous, with the highest rate was reported in Lampung (4.3% mortality and 36.1% hospitalisation), and the lowest rate in East Nusa Tenggara (1.9% mortality and 13.4% hospitalisation). Higher risk of mortality and hospital admission was associated with increased age, clinical diagnosis with pneumonia, presence of pre-existing diabetes, chronic kidney disease, liver disease, and malignancy. Presence of pre-existing hypertension, cardiac diseases, COPD, and immunocompromised conditions were associated with higher risk of hospitalisation, but not associated with mortality. Gender and number of health care workers at province-level were not associated with COVID-19-related mortality and hospitalisation.

The overall mortality rate in these five rural provinces was higher than the mortality rate reported from Jakarta [16] (3.1% vs 1.5%), potentially due to a higher prevalence of at least one comorbidity in these settings (14% vs 0.7%). However, our findings suggest that the hospitalisation rate was slightly lower than in Jakarta [16] (26% vs 33%), possibly due to a lower level of transmission, a lower health care capacity, and a lower access to health care facility in these five provinces. For comparison, as presented in S1 Table, the highest capacity in these settings was seen in Southeast Sulawesi, with 13 doctors per 100,000 population, 37 nurses per 100,000 population, and 13 midwives per 100,000 population. By contrast, Jakarta has 45 doctors per 100,000 population, 48 nurses per 100,000 population, and 31 midwives per 100,000 population. In addition, as previously reported in Jakarta [15], the inequality of COVID-19 vaccine coverage may also contribute to the mortality and hospitalisation rate in Jakarta, but this was not evaluated in the present study. Further study assessing association between health care capacity, including vaccine coverage, with COVID-19-related mortality and hospitalisation is needed to better understand Indonesia's health system resilience in responding the current and future global health crisis.

**Table 3. Mixed effects logistic regression multivariable models showing factors associated with risk of COVID-19 mortality and hospitalisation in five rural provinces in Indonesia, 1 January to 31 July 2021.**

| | Mortality | | Hospitalisation | |
|---|---|---|---|---|
| | aOR (95% CI) | p value | aOR (95% CI) | p value |
| Age group, years | | | | |
| 0–19 | 1 (reference) | | 1 (reference) | |
| 20–29 | 1.5 (0.4–5.8) | 0.537 | 1.1 (0.9–1.5) | 0.354 |
| 30–39 | 1.9 (0.5–6.8) | 0.342 | 1·24 (1.0–1.6) | 0.119 |
| 40–49 | 3.1 (0.9–10.6) | 0.076 | **1.7 (1.3–2.2)** | **<0.001** |
| 50–59 | **6.5 (2.0–21.3)** | **0.002** | **2.9 (2.2–3.8)** | **<0.001** |
| 60–69 | **10.8 (3.2–35.8)** | **<0.001** | **3.3 (2.4–4.4)** | **<0.001** |
| ≥70 | **18.3 (5.3–63.2)** | **<0.001** | **3.7 (2.5–5.5)** | **<0.001** |
| Sex | | | | |
| Female | 1 (reference) | | 1 (reference) | |
| Male | 1.3 (0.9–1.8) | 0.193 | 0.9 (0.8–1.0) | 0.057 |
| Clinical diagnosis with pneumonia | | | | |
| No | 1 (reference) | | 1 (reference) | |
| Yes | **8.0 (5.3–12.0)** | **<0.001** | **15.3 (10.7–21.8)** | **<0.001** |
| Hypertension | | | | |
| No | 1 (reference) | | 1 (reference) | |
| Yes | 1.4 (0.9–2.1) | 0.123 | **2.4 (1.9–3.0)** | **<0.001** |
| Diabetes | | | | |
| No | 1 (reference) | | 1 (reference) | |
| Yes | **2.0 (1.3–3.2)** | **0.002** | **2.2 (1.7–2.9)** | **<0.001** |
| Cardiac diseases | | | | |
| No | 1 (reference) | | 1 (reference) | |
| Yes | 1.6 (0.9–2.9) | 0.119 | **2.1 (1.4–3.1)** | **<0.001** |
| COPD | | | | |
| No | 1 (reference) | | 1 (reference) | |
| Yes | 1.9 (0.7–4.8) | 0.184 | **7.5 (2.3–24.1)** | **0.001** |
| Chronic kidney diseases | | | | |
| No | 1 (reference) | | 1 (reference) | |
| Yes | **6.1 (2.5–14.6)** | **<0.001** | **3.6 (1.6–8.1)** | **0.002** |
| Liver diseases | | | | |
| No | 1 (reference) | | 1 (reference) | |
| Yes | **10.3 (1.8–58.7)** | **0.008** | **7.5 (1.8–31.0)** | **0.006** |
| Malignancy | | | | |
| No | 1 (reference) | | 1 (reference) | |
| Yes | **7.4 (1.5–36.9)** | **0.015** | **3.4 (1.1–10.4)** | **0.029** |
| Immunocompromised | | | | |
| No | 1 (reference) | | 1 (reference) | |
| Yes | ·· | ·· | **6.9 (2.4–20.2)** | **<0.001** |
| Density of doctors | ·· | ·· | ·· | ·· |
| Density of nurses | 1.2 (0.9–1.6) | 0.185 | 0.8 (0.6–1.1) | 0.139 |
| Density of midwives | 1.5 (0.6–3.8) | 0.382 | ·· | ·· |
| Density of public health officers | 1.0 (0.8–1.3) | 0.698 | ·· | ·· |

Province was treated as the random effect variable.

aOR: adjusted odds ratio.

·· the variable did not enter the multivariable model therefore statistics were not estimated.

Health care workers density was calculated as ratio of each type of health workers per 100,000 population

Consistent with a recent finding from an urban area of Indonesia, Jakarta [16], the mortality rate for children under five years old in these rural settings was very low (<1%). This finding was also consistent with several other studies from China [33], Brazil [34, 35], Uganda [17], and South Africa [18] that have reported COVID-19-related deaths among children under 5 years to be rare. Moreover, a review of data from US, Korea and Europe in the early phase of epidemic estimated the overall case fatality rate among children (0–19 years old) to be as low as 0.1% (44/42 846) [36].

Our findings suggest that COVID-19-related mortality and hospitalisation rates varied across the five provinces. For example, compared to in East Nusa Tenggara, the rate in Lampung was 2.3 times (4.3% vs 1.9%) for mortality and 2.7 times (36.1% vs 13.4%) for hospitalisation. The observed different risks of COVID-19-related mortality and hospitalisation could be due to several factors such as different prevalence of older and comorbid population, as well as different level of health care capacity across the provinces. Although our study found an absence of association between number of health care workers at province-level with COVID-19-related mortality and hospitalisation, a previous nationwide ecological study in Indonesia, suggested that COVID-19 has disproportionately affected district with high proportion of elder population, high prevalence of diabetes mellitus, lower doctor to population ratio, higher life expectancy at birth, and lower level of formal education [37]. Other health care factors such as quality of COVID-19 diagnosis and case management may also affect the COVID-19 heterogeneity, but these factors were not evaluated in the current study.

Risk of mortality and hospitalisation increased with the presence of one or more comorbidities, especially pre-existing diabetes, chronic kidney diseases, liver diseases, or malignancy, and clinical pneumonia. Based on Table 3, the mortality risk increased by 2.0-fold, 6.1-fold, 10.3-fold, 7.4-fold, and 8.0-fold for people with pre-existing diabetes, chronic kidney diseases, liver diseases, malignancy, and a clinical diagnosis with pneumonia, respectively. The presence of pre-existing hypertension, cardiac diseases, COPD, or immunocompromised conditions also increased risk of hospitalisation by 2.4-fold, 2.1-fold, 7.5-fold, and 6.9-fold, respectively, but was not associated with increased risk of death. In addition, mortality and hospitalisation risk were increased for people which were clinically diagnosed with pneumonia. These findings are consistent with evidence from previous studies suggesting that pre-existing comorbidities, including cardiovascular or cerebrovascular diseases, hypertension, and diabetes mellitus, were associated with poorer COVID-19 outcomes [6–8, 38]. Although chronic comorbidities among COVID-19 cases in these five rural provinces (14.3%) were less frequent than reported in more developed settings including North America [6, 7], South America [27], and Europe [8], this finding could reflect the under-reporting of comorbidities by patients, or under-diagnosis due to variable access to quality health services. Finally, obesity has also been recognised as an important risk factor of mortality [8, 19], but we could not assess this due to absence of data for this variable.

This study has limitations. The retrospective design and reliance on routine surveillance data meant that data were incomplete or unavailable for some key baseline variables (e.g., date of symptom onset, date of hospital admission and outcome, vital signs, previous SARS-CoV-2 infection, and disease severity classification). As the national COVID-19 vaccination program has just started in the middle of January 2021 and in 14 selected high-priority provinces, excluding our study settings, we were not able to assess the vaccine effectiveness in the present study. The imperfect contact tracing, testing, and reporting activities could result in underreporting of cases, especially for asymptomatic and mild cases which thus results in overestimation of rates of mortality and hospitalisation rate. Comorbidities were often self-reported or could be under-diagnosed, potentially resulting in underreporting and hence underestimation of effect sizes. Findings from this study may not reflect the mortality and hospitalisation rate,

and risk factors associated with COVID-19-related outcomes in other LMIC settings with different demographics, non-infectious diseases prevalence, and different health care capacity.

## Conclusion

In conclusion, risk factors associated with severe COVID-19 outcomes in five rural provinces in Indonesia are dominated by advanced age and pre-existing comorbidities. Heterogeneity of mortality and hospitalisation rate in the studied areas was likely driven by the different age structure and prevalence of pre-existing comorbidities. The findings highlight the need for enhanced context-specific public health action to reduce mortality and hospitalisation risk among older and comorbid rural populations, and improved care and treatment for the cases. Although health care worker capacity was not associated with mortality and hospitalisation in this study, future nationwide study assessing association between health care capacity and COVID-19 impacts is needed to better guide improvement of health systems resilience in the future.

## Supporting information

**S1 Fig. Number of COVID-19 outcomes over time by province.**
(TIF)

**S1 Table. Population number and number of health care workers by province.**
(DOCX)

**S2 Table. Number of cases, population number and COVID-19 incidence rate by province.**
(DOCX)

**S3 Table. Bivariable mixed effects logistic regression analysis showing factors associated with risk of COVID-19 mortality and hospitalisation in five rural provinces, Indonesia.**
(DOCX)

**S4 Table. Mixed effects logistic regression multivariable models assessing association between and mortality and hospitalisation with number of comorbidities in five rural provinces, Indonesia.**
(DOCX)

**S1 Checklist.**
(DOCX)

## Acknowledgments

We acknowledge all health care workers involved in the care for the COVID-19 patients, as well as those involved in the field data collection.

## Author Contributions

**Conceptualization:** Henry Surendra.

**Data curation:** Henry Surendra, Arina M. Ersanti, Mariati Rahmat, Widia Noviyanti, Erni N. A. Mansur, Yana Y. Suleman, Sitti Sudrani, Rosalina Rosalina, Ismen Mukhtar.

**Formal analysis:** Henry Surendra.

**Funding acquisition:** William A. Hawley, Hariadi Wibisono.

**Investigation:** Henry Surendra.

**Methodology:** Henry Surendra.

**Project administration:** Dian Rosadi, Lukman Fauzi.

**Supervision:** Henry Surendra, C. Yekti Praptiningsih, Hariadi Wibisono.

**Validation:** Henry Surendra, Arina M. Ersanti, Mariati Rahmat, Widia Noviyanti, Joshua A. D. Harmani, Erni N. A. Mansur, Yana Y. Suleman, Sitti Sudrani, Rosalina Rosalina, Ismen Mukhtar.

**Visualization:** Henry Surendra.

**Writing – original draft:** Henry Surendra, C. Yekti Praptiningsih, Iqbal R. F. Elyazar, William A. Hawley.

**Writing – review & editing:** Henry Surendra, C. Yekti Praptiningsih, Arina M. Ersanti, Mariati Rahmat, Widia Noviyanti, Joshua A. D. Harmani, Erni N. A. Mansur, Yana Y. Suleman, Sitti Sudrani, Rosalina Rosalina, Ismen Mukhtar, Dian Rosadi, Lukman Fauzi, Iqbal R. F. Elyazar, William A. Hawley, Hariadi Wibisono.

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
