## [Decision Letter · Decision Letter 0]

12 Dec 2022

PONE-D-22-31666Clinical characteristics and factors associated with COVID-19-related mortality and hospital admission in 5 rural provinces in Indonesia: a retrospective cohort studyPLOS ONE

Dear Dr. Surendra,

Thank you for submitting your manuscript to PLOS ONE. After careful consideration, we feel that it has merit but does not fully meet PLOS ONE’s publication criteria as it currently stands. Therefore, we invite you to submit a revised version of the manuscript that addresses the points raised during the review process.

We look forward to receiving your revised manuscript.

Kind regards,

Harapan Harapan, MD, PhD

Academic Editor

PLOS ONE

Additional Editor Comments:

This study have been assessed by three reviewers with slightly different views. After reading the manuscript carefully, this manuscript is potentially to be improved by addressing issues raised by the reviewer. Please address all issues concerned by the reviewers and revise the manuscript as requested or provide the explanations to their questions. During your revision, Lines 84-85 please briefly touch on problem of less effective vaccine on new strains of SARS-CoV-2 and limited approved anti-viruses (see 10.52225/narra.v2i3.88 - 10.52225/narra.v2i3.92).

Reviewers' comments:

Reviewer's Responses to Questions

**Comments to the Author**

1. Is the manuscript technically sound, and do the data support the conclusions?

Reviewer #1: No

Reviewer #2: Partly

Reviewer #3: Yes

2. Has the statistical analysis been performed appropriately and rigorously? 

Reviewer #1: No

Reviewer #2: No

Reviewer #3: Yes

3. Have the authors made all data underlying the findings in their manuscript fully available?

Reviewer #1: No

Reviewer #2: No

Reviewer #3: Yes

4. Is the manuscript presented in an intelligible fashion and written in standard English?

Reviewer #1: Yes

Reviewer #2: Yes

Reviewer #3: Yes

5. Review Comments to the Author

Reviewer #1: 1. COVID-19 confirmation is not solely based on PCR. COVID-19 cases from antigen tests should be excluded considering the inaccuracy of the method (especially with variants of concern emergence).

2. Not sure about the immune status of the patients (i.e. previous infection history or vaccination).

3. Difference in the number of symptomatic patients in deceased group is suspicious. This could probably due to the disease progression which could be different during the initial admission. Hence, it is not accurate to include clinical feature in the model.

4. Usually the “Ethics approval; Consent for publication; Availability of data and materials; Conflict of interests; and Funding” are placed after ‘Conclusion’.

5. Line 134—137 not sure what the percentages are for.

Reviewer #2: Thank you for the opportunity to review this article. This is an intriguing study. However, some parts of the manuscript still require improvement. I provide some questions and suggestions that I think will improve the quality of the manuscript. Please read more specific comments in the attached file.

Reviewer #3: thnaks for the opportunity to review. Very well presented data - straightforward analysis. Limitations acknowledged especially as regards to making comparisons about mortality with likely uncertain case finding/denominator.

6. PLOS authors have the option to publish the peer review history of their article (what does this mean?). If published, this will include your full peer review and any attached files.

Reviewer #1: No

Reviewer #2: No

Reviewer #3: No

---

## [Author Response · Author response to Decision Letter 0]

24 Jan 2023

Point by point response to reviewers’ comments

Editor comments:

Author response: We have now adjusted our manuscript to meet the PLOS ONE style.

Author response: We thank the editor for pointing this out. We have now added the correct funding information.

Author response: The data contain de-identified dataset own by the Indonesian Epidemiologist Association. We have now included the following statement in the manuscript and the cover letter:

“The data contain de-identified data set own by the Indonesian Epidemiologist Association/Perhimpunan Ahli Epidemiologi Indonesia. After publication, the datasets used for this study will be made available to others on reasonable requests to the corresponding author and the Indonesian Epidemiologist Association (email: info@paei.or.id), including a detailed research proposal, study objectives and statistical analysis plan.”

Author response: We thank the editor for providing the guidelines. We would like to clarify that the shapefiles map used in Figure 1 was obtained from the official website of Indonesia Geospatial Agency. As stated in the website, the use and distribution of the map is permitted, provided we acknowledge the copyright holder (https://tanahair.indonesia.go.id/portal-web/termcondition). This map has been used in many other scientific publications, including in the following COVID-19 paper published by PLOS ONE (https://journals.plos.org/plosone/article/figure?id=10.1371/journal.pone.0243703.g003). We have now acknowledged the copyright holder in the caption of Figure 1.

Author response: We have now included the captions for our Supporting Information files and updated the in-text citations accordingly.

Additional Editor Comments:

This study have been assessed by three reviewers with slightly different views. After reading the manuscript carefully, this manuscript is potentially to be improved by addressing issues raised by the reviewer. Please address all issues concerned by the reviewers and revise the manuscript as requested or provide the explanations to their questions. During your revision, Lines 84-85 please briefly touch on problem of less effective vaccine on new strains of SARS-CoV-2 and limited approved anti-viruses (see 10.52225/narra.v2i3.88 - 10.52225/narra.v2i3.92).

Author response: We thank the editor for their very positive feedback. We have now revised and responded to all comments from reviewers, and have also briefly touched on problem of less effective vaccine on new strains of SARS-CoV-2 and limited approved anti-viruses (Line 78-79). Please find the point-by-point response to reviewer comment below:

Reviewer #1: 

Reviewer comment #1

1. COVID-19 confirmation is not solely based on PCR. COVID-19 cases from antigen tests should be excluded considering the inaccuracy of the method (especially with variants of concern emergence).

Author response: We thank the reviewer for this comment. Although we acknowledge the limitation of some AgRDTs especially in the emergence of new SARS-CoV-2 variants, we would argue that the case definition used in our study was based on the official definition used by the Indonesian government, which was based on recommendation from the WHO (https://www.who.int/indonesia/news/detail/08-12-2020-implementation-of-antigen-rdt-(ag-rdt)-to-detect-covid-19-cases-in-indonesia). The Indonesian government has been following the WHO recommendation on the use of AgRDTs with ≥ 80% sensitivity and ≥ 97-100% specificity compared to an NAAT reference assay. For that reason, we believe that the inclusion of AgRDT-confirmed cases was methodologically acceptable.

Reviewer comment #2

2. Not sure about the immune status of the patients (i.e. previous infection history or vaccination).

Author response: We acknowledge the important role of previous infection and history of vaccination in explaining risk of COVID-19 mortality. However, the data on history of infection were not available for our analysis. Moreover, the vaccination program has only been started in mid of January 2021 in 14 selected high-priority provinces, excluding our study settings. We believe that the effect of vaccination in reducing mortality could not be seen in our study period (1 January to 31 July 2021), considering the gap in vaccine rollout, low vaccination rate outside Java Island, and the time gap between the first and second dose and time to reach optimum protection from the two-dose vaccines.

Reviewer comment #3

3. Difference in the number of symptomatic patients in deceased group is suspicious. This could probably due to the disease progression which could be different during the initial admission. Hence, it is not accurate to include clinical feature in the model.

Author response: We thank the reviewer for this comment and would like to clarify that we did not include the clinical features (i.e., sign and symptoms) in the regression models. The clinical features were only assessed in descriptive analysis in Table 1. The clinical characteristics included in the regression models were clinical pneumonia and type of pre-existing comorbidities (Table 3).

Reviewer comment #4

4. Usually the “Ethics approval; Consent for publication; Availability of data and materials; Conflict of interests; and Funding” are placed after ‘Conclusion’.

Author response: Thank you for pointing this out. We have now revised the manuscript structure accordingly.

Reviewer comment #5

5. Line 134—137 not sure what the percentages are for.

Author response: The percentages are referring to the proportion of areas (villages) within each province that were categorised as rural areas by the government. We have now added “villages” term to make the statement clearer:

“Based on data from Indonesia Central Bureau of Statistics, the majority of areas (villages) in the five provinces were categorised as rural areas i.e., Lampung (68%), Gorontalo (66%), Central Sulawesi (55%), Southeast Sulawesi (83%), and East Nusa Tenggara (92%)” Line (140-143).

Reviewer 2

Thank you for the opportunity to review this article. This is an intriguing study. However, some parts of the manuscript still require improvement. I provide some questions and suggestions that I think will improve the quality of the manuscript. Please read more specific comments in the attached file.

Reviewer comment #1

Abstract

Methods: I suggest the authors to include brief information about SISUGI information system in the abstract because this research is part of the evaluation of the program.

Author response: We thank the reviewer for highlighting this information. We have now added the following information about SISUGI:

“We extracted demographic and clinical data, including hospitalisation and mortality from a new piloted COVID-19 information system named Sistem Informasi Surveilans Epidemiologi (SISUGI).” (Abstract, Methods Line 45-48)

Reviewer comment #2

L49-55: It will be more accurate if it uses one decimal place.

Author response: We have now revised as suggested.

Reviewer comment #3

Line 61: I thought they were dissimilar. Moreover, this conclusion should be supported by more data/information (especially that presented in the discussion section). 

Author response: We thank the reviewer for this comment and would like to clarify that what we meant here was the similarity in terms of the risk factors mentioned in the next sentence. For clarity, we have now deleted that statement.

Reviewer comment #4

Please add a brief recommendation to the Indonesian government regarding the findings in this study. 

Author response: We have now added the following recommendation:

“The findings highlight the need for prioritising enhanced context-specific public health action to reduce mortality and hospitalisation risk among older and comorbid rural populations.” (Abstract, Line 66-68)

Reviewer comment #5

Background, Line 113: The Indonesian government has developed several applications/systems to deal with Covid-19 in Indonesia like PeduliLindungi, etc. Please elaborate on this issue in this paragraph.

Author response: We thank the reviewer for the suggestion. We have now added the following sentences to describe this nuance:

“The Indonesian government has been implementing a national COVID-19 data collection system (New All Record) that collect limited demographic data (sex, age, and residential) of individuals tested positive by either polymerase chain reaction (PCR) or rapid antigen diagnostic test (Ag RDT) [20,21], followed by the Peduli Lindungi [22], a system that records data on vaccination status, latest PCR/Ag RDT results, and travel history. To improve quality of COVID-19 surveillance and response in Indonesia, the Indonesian Epidemiologist Association (Perhimpunan Ahli Epidemiologi Indonesia, PAEI) in collaboration with the Ministry of Health of Indonesia implemented a new COVID-19 information system named Sistem Informasi Surveilans Epidemiologi (SISUGI) [24] to collect more detailed clinical and epidemiological data in five rural provinces – East Nusa Tenggara in eastern Indonesia, Gorontalo, Central Sulawesi, and Southeast Sulawesi on the island of Sulawesi, and Lampung on the island of Sumatra” (Line 107-119)

Reviewer comment #6

Background, Line 122: Is there a specific reason why this time period was chosen because the development of the Covid-19 cases is very dynamic from time to time?

Author response: We thank the reviewer for this comment. We acknowledge that the COVID-19 transmission is very dynamic over time. However, as previously mentioned in Methods section, this study was part of the implementation of SISUGI pilot program. The program was implemented during the period of January to July 2021; thus, we were only able to utilised and analysed data collected during this time period. There was no other specific scientific reason of choosing this time period.

Reviewer comment #7

Methods, Line 160: Please add the data source.

Author response: We have now cited the data source.

Reviewer comment #8

Methods, Line 164: Please elaborate why the author did not employ a parametric test. Furthermore, in which tables were the results of the Mann-Whitney U test presented?

Author response: We thank the reviewer for this comment. We used Mann-Whitney U test to assess median different of age between deceased and recovered cases, and between hospitalised and non-hospitalised cases. The results of the test were presented in Table 1 “Median age” variable. The age distribution does not follow normal distribution, therefore non-parametric test was used instead of the parametric test. For clarity, we have now revised the statement as follows:

“We used the Mann-Whitney U test to compare median different of age between deceased and recovered cases, and between hospitalised and non-hospitalised cases. For categorical variables, we used Chi-squared test, or Fisher’s exact test to compare other characteristics between deceased and recovered cases, and between hospitalised and non-hospitalised cases.” (Statistical analysis Line 169-173)

Reviewer comment #9

Line 173: This step of analysis is not clear to me.

Author response: We thank the reviewer for this comment. This first step is a common first step to check the appropriateness of the multi-level model (i.e., mixed effect logistic regression in this case). The null model provides the intraclass correlation coefficient (ICC) estimate which can be used to justify whether a mixed model is even necessary: an ICC of zero (or very close to zero) means the observations within clusters (province) are no more similar than observations from different clusters, and setting it as a random factor might not be necessary. The ICC was 5% for mortality and 18% for hospitalisation, thus, strongly justify the use of mixed effect models treating province as the random effect variable to adjust for clustering of observations within provinces.

Reviewer comment #10

Line 180: “along with other demographic and clinical factors”

I think these factors were analyzed in the first model (supplementary Table 3), not in the second model.

Author response: We thank the reviewer for this comment. We would like to clarify several things. Firstly, the first and the second model mentioned in this section was referring to the two final multivariable models (as mentioned in Line 176) presented in Table 3 and Supplementary Table 4. Secondly, the different between the first and the second model is on the way we treat the comorbidity data. In the first final model we assess the effect of each type of comorbidities (hypertension, diabetes, etc.), along with other demographic and clinical factors. In the second model, we focus our analysis to see the effect of the number of comorbidities on death and hospitalisation, controlling for other demographic and clinical factors. Indeed, this means that we included the demographic and clinical factors in both multivariable models.

For more clarity, we have now revised the statements as follow:

“We reported two final multivariable logistic regression models. In the first model, we assessed the effect of each type of comorbidity (hypertension, diabetes, cardiac disease, chronic obstructive pulmonary disease (COPD), chronic kidney disease, immunocompromised status, and liver disease), along with other demographic and clinical factors. In the second model, we assessed the effect of the number of comorbidities (0, 1 or >1), controlling for other demographic and clinical factors.” (Line 184-190)

Reviewer comment #11

Line 180: “interaction terms”

There was no interaction term between age and sex found in the analysis

Author response: This statement reports that we used interaction terms to examine potential effect modification by age and sex. However, we found no interactions in our analysis, therefore, the interaction term was not included in the final model. We have indicated that sex was not found to be an effect modifier in either model, but missed to mention age. We have now revised the statement to make it clearer:

“Sex and age were not found to be an effect modifier in either model” (Results Line 298)

Reviewer comment #12

Line 184: Where can I find this STROBE checklist? I suggest the authors include the checklist as a supplementary file.

Author response: We have now included the checklist as supplementary.

Reviewer comment #13

Results Line 213: The number of cases in the five provinces appears to be low for that time period. Have the authors compared it with official data released by the government, for example via www.covid19.go.id? 

Author response: We thank the reviewer for this comment. We acknowledge that the number of cases recorded in the SISUGI were likely to be lower than the total numbers of cases reported to the national COVID-19 database (New All Record/NAR). This is mostly because the different in source of data used in these two databases. While SISUGI was based on data captured by public health facilities, the total number of cases recorded in the NAR were based on COVID-19 PCR test or AgRDT testing data from all public and private laboratories. We have acknowledged the limitation of using routine surveillance data in Discussion, Line 377. Moreover, as reported by the Ministry of Health Indonesia on 11 August 2021, the data recorded in the national database were not real time because there was a substantial reporting delay (https://www.kemkes.go.id/article/view/21081200001/lonjakan-angka-kematian-covid-19-akibat-akumulasi-kasus-yang-belum-terlaporkan.html). Therefore, we were not able to make a reliable comparison between the data recorded in SISUGI with those recorded in the national database.

Reviewer comment #14

Results Line 229-235: These data are also related to Table 1

Author response: We have now also cited Table 1

Reviewer comment #15

Results Line 239: 34.2+26.3=60.5%? (Table 1) and 10.0+2.9=12.9%? (Table 1)

Line 245: 25.0+9.9=34.9%? (Table 1) and 5.7+1.4=7.1%? (Table 1)

Author response: We thank the reviewer for pointing out this error. We have now revised as per the actual value on Table 1.

Reviewer comment #16

Results Table 1: I think it would be more comprehensible if the percentage for deceased and hospitalized were calculated by row instead of the column. For example for deceased in Lampung (yes=93 (4.3%) and no=2,053 (95.7%)). 

Author response: We thank the reviewer for this comment. As the purpose of the information is to describe the proportion of cases by province as mentioned in Results Line 206-207, we choose to keep the current presentation and make it consistent with the presentation of other variables summarised in Table 1. Moreover, we have already presented the suggested province-specific mortality and hospitalisation rate in Table 2, and highlighted the following sentence in Line 247-250:

“Based on province, the mortality and hospitalisation rate ranged from 1.9% (13/658) and 13.4% (88/658) in East Nusa Tenggara to 4.3% (93/2,146) and 36.1% (775/2,146) in Lampung. Details on province and age-specific mortality and hospitalisation rate can be seen in Table 2.”

Reviewer comment #17

Results Line 255: Based on Table 1 and Table 2, it should be 215.

Author response: Thank you for pointing out this error. It is now revised based on the actual value on Table 1 and 2.

Reviewer comment #18

Results Line 268: Please indicate all categories defined as references in bivariate and multivariate logistic regression analysis. For example, it is unclear which category for hypertension is set as a reference (yes or no). For other variables like age groups or gender, the reference category is clearly stated in the table.

Author response: We thank the reviewer for this comment. We have now added reference for each categorical variables assessed in the regression.

Reviewer comment #19

Results Line 269: How about gender? I think it was significant in bivariate analysis (supplementary Table 3).

Author response: The reviewer is correct, higher mortality risk was associated with male sex in bivariable analysis but missing from the text. We have now added it to the list.

“In bivariable mixed effects logistic regression analysis (S3 Table), the risk of death was associated with older age, male sex, clinical diagnosis with pneumonia, number of pre-existing comorbidities, hypertension, diabetes, cardiac disease, COPD, chronic kidney disease, liver diseases, malignancy, lower number of nurses per 100,000 population, and higher number of midwives per 100,000 population.” (Line 260-264)

Reviewer comment #20

Results Line 270: No results for COPD regarding mortality were presented in supplementary Table 3.

Author response: We thank the reviewer for spotting this out. We have now added the statistics for COPD in Bivariable analysis in S3 Table.

Reviewer comment #21

Results Table 3: In the bivariate analysis, "number of comorbidities" was a significant factor (p-value < 0.05) for mortality and hospitalisation. However, this variable was not included in the multivariate model. 

Author response: The reviewer is correct; the number of comorbidities was significantly associated in bivariable. As mentioned in Methods Line 184-190, we reported two final multivariable models where we reported the effect of each type of comorbidities in the first model (Results, Table 3), and the effect of number of comorbidities in the second model (S4 Table).

Reviewer comment #22

Results Table 3: Sex was not a significant factor for hospitalisation (p value = 0.623), but it was included in multivariate analysis. This is contrary to what the authors have stated in the statistical analysis subsection.

Author response: The reviewer is correct that sex was not statistically associated with hospitalisation. However, sex has been known to be a strong predictor of not only the dependent variables, but also for the other independent variables (e.g., different type of comorbidities) assessed in this study. Therefore, we treated sex as a “forced variable” that must be controlled in the multivariable models. We have now added the following statement in Methods section to make it clearer and consistent:

“Sex was treated as a “forced variable” (a strong potential confounder to be included in multivariable models, regardless of its p-value in bivariable analysis).” (Methods Line 182-184)

Reviewer comment #23

Results Line 289: 58.7 based on Table 3.

Results Line 297: 24.1 based on Table 3.

Author response: Thank you for spotting the errors. We have now revised it based on the actual value on Table 3.

Reviewer comment #24

Discussion Line 317: It should be 215.

Author response: It is now revised based on the actual value presented in Table 1.

Reviewer comment #25

Discussion Line 328-330: Please include the source of these data.

Author response: We have now cited the reference.

Reviewer comment #26

Discussion Line 349: These statistics were unclear to me

Author response: These statistics were the estimates of the overall COVID-19 case fatality rate among children (0–19 years old) reported by a review conducted in the early phase of the pandemic (https://www.sciencedirect.com/science/article/pii/S0033350620302092?via%3Dihub). The 0.03 per 100,000 was referring to the number of deaths per 100,000 of 0-19 years old population (standardised estimate), while the (44/42,846) was referring to the number of deaths per total cases. For clarity, we have now revised the statistics to only include the case fatality rate as follows:

“Moreover, a review of data from US, Korea and Europe in the early phase of epidemic estimated the overall case fatality rate among children (0–19 years old) to be as low as 0.1% (44/42 846)” (Line 340-342)

Reviewer comment #27

Discussion Line 366 and 370: It is unclear to me. Please elaborate how these percentages were calculated. From which table were these data obtained?

Author response: We thank the reviewer for this comment. These data are the relative risk different transformed from the estimated adjusted Odds Ratio presented in Table 3. For example, the mortality risk increased by 102% (aOR 2.02), 505% (aOR 6.05) for people with pre-existing diabetes and chronic kidney diseases, respectively. However, for clarity we have now revised the statements as follows:

“Risk of mortality and hospitalisation increased with the presence of one or more comorbidities, especially pre-existing diabetes, chronic kidney diseases, liver diseases, or malignancy, and clinical pneumonia. Based on Table 3, the mortality risk increased by 2.0-fold, 6.1-fold, 10.3-fold, 7.4-fold, and 8.0-fold for people with pre-existing diabetes, chronic kidney diseases, liver diseases, malignancy, and a clinical diagnosis with pneumonia, respectively. The presence of pre-existing hypertension, cardiac diseases, COPD, or immunocompromised conditions also increased risk of hospitalisation by 2.4-fold, 2.1-fold, 7.5-fold, and 6.9-fold, respectively, but was not associated with increased risk of death. In addition, mortality and hospitalisation risk were increased for people which were clinically diagnosed with pneumonia.” (Line 358-365)

Reviewer comment #28

Conclusion Line 395: I thought they were dissimilar. Moreover, this conclusion should be supported by more data/information. 

Author response: We thank the reviewer for this comment and would like to clarify that what we meant here was the similarity in terms of the clinical and demographic risk factors mentioned in the next sentence. As the main objective of this study was to examine those risk factors associated with COVID-19 mortality and hospitalisation in rural settings of Indonesia, we have now revised our conclusion statement as follows:

“In conclusion, risk factors associated with severe COVID-19 outcomes in five rural provinces in Indonesia are dominated by advanced age and pre-existing comorbidities. Heterogeneity of mortality and hospitalisation rate in the studied areas was likely driven by the different age structure and prevalence of pre-existing comorbidities. The findings highlight the need for enhanced context-specific public health action to reduce mortality and hospitalisation risk among older and comorbid rural populations, and improved care and treatment for the cases. Although health care worker capacity was not associated with mortality and hospitalisation in this study, future nationwide study assessing association between health care capacity and COVID-19 impacts is needed to better guide improvement of health systems resilience in the future.”

Reviewer 3

Reviewer #3: thanks for the opportunity to review. Very well presented data - straightforward analysis. Limitations acknowledged especially as regards to making comparisons about mortality with likely uncertain case finding/denominator.

Author response: We thank the reviewer for their very positive comment.

---

## [Decision Letter · Decision Letter 1]

8 Mar 2023

PONE-D-22-31666R1Clinical characteristics and factors associated with COVID-19-related mortality and hospital admission in 5 rural provinces in Indonesia: a retrospective cohort studyPLOS ONE

Dear Dr. Surendra,

Thank you for submitting your manuscript to PLOS ONE. After careful consideration, we feel that it has merit but does not fully meet PLOS ONE’s publication criteria as it currently stands. Therefore, we invite you to submit a revised version of the manuscript that addresses the points raised during the review process.

We look forward to receiving your revised manuscript.

Kind regards,

Harapan Harapan, MD, PhD

Academic Editor

PLOS ONE

Journal Requirements:

Reviewers' comments:

Reviewer's Responses to Questions

**Comments to the Author**

1. If the authors have adequately addressed your comments raised in a previous round of review and you feel that this manuscript is now acceptable for publication, you may indicate that here to bypass the “Comments to the Author” section, enter your conflict of interest statement in the “Confidential to Editor” section, and submit your "Accept" recommendation.

Reviewer #1: All comments have been addressed

Reviewer #2: (No Response)

2. Is the manuscript technically sound, and do the data support the conclusions?

Reviewer #1: Yes

Reviewer #2: Yes

3. Has the statistical analysis been performed appropriately and rigorously? 

Reviewer #1: Yes

Reviewer #2: Yes

4. Have the authors made all data underlying the findings in their manuscript fully available?

Reviewer #1: No

Reviewer #2: No

5. Is the manuscript presented in an intelligible fashion and written in standard English?

Reviewer #1: Yes

Reviewer #2: Yes

6. Review Comments to the Author

Reviewer #1: Authors have addressed all my concerns. Some minor comments were left, but the manuscript does not need to be sent back to me.

Query #2: Regarding immune status from previous infection. Please clarify your statement in the manuscript.

Additional comments:

1. The data were form “1 January to 31 July 2021”. For such data the title is too general. Kindly state which epidemic wave the data were from.

2. It is 2023 already, while your data were from 2021. Please highlight the importance of having these data.

Reviewer #2: Thank you to the authors for their effort to revise the manuscript. From my point of view, the manuscript is getting better and clearer after revision. However, I found that two comments from my previous review were not sufficiently addressed.

1). I can not find the STROBE checklist in the attachments file as mentioned by the authors. Please clarify this issue.

2). Commonly, a p-value is written as “<0.001" instead of "<0.0001". (Table 1, Table 3, S3 Table & S4 Table).

7. PLOS authors have the option to publish the peer review history of their article (what does this mean?). If published, this will include your full peer review and any attached files.

Reviewer #1: No

Reviewer #2: No

---

## [Author Response · Author response to Decision Letter 1]

15 Mar 2023

Point by point response to reviewers’ comments

Editor comments:

Author response: We thank the editor for their very positive feedback. We have reviewed the accuracy and completeness of our reference list and confirm that there are no retracted papers cited in this manuscript. Please find the point-by-point response to reviewer comment below:

Reviewers' comments:

Reviewer's Responses to Questions

Comments to the Author

1. If the authors have adequately addressed your comments raised in a previous round of review and you feel that this manuscript is now acceptable for publication, you may indicate that here to bypass the “Comments to the Author” section, enter your conflict of interest statement in the “Confidential to Editor” section, and submit your "Accept" recommendation.

Reviewer #1: All comments have been addressed

Reviewer #2: (No Response)

2. Is the manuscript technically sound, and do the data support the conclusions?

Reviewer #1: Yes

Reviewer #2: Yes

3. Has the statistical analysis been performed appropriately and rigorously?

Reviewer #1: Yes

Reviewer #2: Yes

4. Have the authors made all data underlying the findings in their manuscript fully available?

Reviewer #1: No

Reviewer #2: No

Author response: As previously requested by editor, we have stated the following statemen in our cover letter:

“We would like to also specifically state that the data analysed in this study contain de-identified dataset own by the Indonesian Epidemiologist Association/Perhimpunan Ahli Epidemiologi Indonesia. After publication, the datasets used for this study will be made available to others on reasonable requests to the corresponding author and the Indonesian Epidemiologist Association (email: info@paei.or.id), including a detailed research proposal, study objectives and statistical analysis plan.”

5. Is the manuscript presented in an intelligible fashion and written in standard English?

Reviewer #1: Yes

Reviewer #2: Yes

Reviewer #1: 

Authors have addressed all my concerns. Some minor comments were left, but the manuscript does not need to be sent back to me.

Reviewer comment #1

Query #2: Regarding immune status from previous infection. Please clarify your statement in the manuscript.

Author response: We thank the reviewer for this comment. We have now added the following statement in Limitations, Discussion Line 383-389:

“This study has limitations. The retrospective design and reliance on routine surveillance data meant that data were incomplete or unavailable for some key baseline variables (e.g., date of symptom onset, date of hospital admission and outcome, vital signs, previous SARS-CoV-2 infection, and disease severity classification). As the national COVID-19 vaccination program has just started in the middle of January 2021 and in 14 selected high-priority provinces, excluding our study settings, we were not able to assess the vaccine effectiveness in the present study.”

Reviewer comment #2

Additional comments:

1. The data were form “1 January to 31 July 2021”. For such data the title is too general. Kindly state which epidemic wave the data were from.

Author response: We thank the reviewer for this comment. We have now emphasised that the data were from the first two epidemic waves in Indonesia. We have stated this in the title to make it more precise and accurate, and have also added the following sentence to give more nuance of the first two epidemic waves occurred in Indonesia:

“The first epidemic wave occurred from 2 March 2020 to 30 April 2021, and the more intense second wave dominated by Delta variant peaked in July 2021” (Background, Line 98-99)

Reviewer comment #3

2. It is 2023 already, while your data were from 2021. Please highlight the importance of having these data.

Author response: We thank the reviewer for this comment. We have now added the following sentences to highlight the importance of having these data:

“The study findings will inform decision on prioritising clinical and public health interventions to reduce risk of hospital admission and mortality associated with COVID-19 in other rural settings of Indonesia.” (Background, Line 123-125)

Reviewer #2: 

Reviewer comment #1

Thank you to the authors for their effort to revise the manuscript. From my point of view, the manuscript is getting better and clearer after revision. However, I found that two comments from my previous review were not sufficiently addressed.

1). I can not find the STROBE checklist in the attachments file as mentioned by the authors. Please clarify this issue.

Author response: We thank the reviewer for their positive comments. We can confirm that we have submitted the STROBE checklist in the previous round. However, it seems that the checklist is only for editor review and was not sent out to the reviewers. 

Reviewer comment #2

2). Commonly, a p-value is written as “<0.001" instead of "<0.0001". (Table 1, Table 3, S3 Table & S4 Table).

Author response: We thank the reviewer for this comment and apologise for the errors. We have now revised it as “<0.001" accordingly.

---

## [Editor Report · Decision Letter 2]

20 Mar 2023

Clinical characteristics and factors associated with COVID-19-related mortality and hospital admission during the first two epidemic waves in 5 rural provinces in Indonesia: a retrospective cohort study

PONE-D-22-31666R2

Dear Dr. Surendra,

We’re pleased to inform you that your manuscript has been judged scientifically suitable for publication and will be formally accepted for publication once it meets all outstanding technical requirements.

Kind regards,

Harapan Harapan, MD, PhD

Academic Editor

PLOS ONE
---

## [Editor Report · Acceptance letter]

22 Mar 2023

PONE-D-22-31666R2 

Clinical characteristics and factors associated with COVID-19-related mortality and hospital admission during the first two epidemic waves in 5 rural provinces in Indonesia: a retrospective cohort study 

Dear Dr. Surendra:

I'm pleased to inform you that your manuscript has been deemed suitable for publication in PLOS ONE. Congratulations! Your manuscript is now with our production department. 

Kind regards, 

on behalf of

Dr. Harapan Harapan 

Academic Editor

PLOS ONE